# Red Blood Cell Membrane-Camouflaged Tedizolid Phosphate-Loaded PLGA Nanoparticles for Bacterial-Infection Therapy

**DOI:** 10.3390/pharmaceutics13010099

**Published:** 2021-01-14

**Authors:** Xinyi Wu, Yichen Li, Faisal Raza, Xuerui Wang, Shulei Zhang, Ruonan Rong, Mingfeng Qiu, Jing Su

**Affiliations:** School of Pharmacy, Shanghai Jiao Tong University, Shanghai 200240, China; xinyi.woo@sjtu.edu.cn (X.W.); liyichen592@sjtu.edu.cn (Y.L.); faisalraza@sjtu.edu.cn (F.R.); sherryw33@sjtu.en (X.W.); sjtu.zsl@sjtu.edu.cn (S.Z.); ruonanrong@sjtu.edu.cn (R.R.)

**Keywords:** tedizolid phosphate (TR-701), red blood cell membrane (RBCM), methicillin-resistant staphylococcus aureus (MRSA), infection, exotoxins removing

## Abstract

Multiple drug resistance (MDR) in bacterial infections is developed with the abuse of antibiotics, posing a severe threat to global health. Tedizolid phosphate (TR-701) is an efficient prodrug of tedizolid (TR-700) against gram-positive bacteria, including methicillin-sensitive staphylococcus aureus (MSSA) and methicillin-resistant staphylococcus aureus (MRSA). Herein, a novel drug delivery system: Red blood cell membrane (RBCM) coated TR-701-loaded polylactic acid-glycolic acid copolymer (PLGA) nanoparticles (RBCM-PLGA-TR-701NPs, RPTR-701Ns) was proposed. The RPTR-701Ns possessed a double-layer core-shell structure with 192.50 ± 5.85 nm in size, an average encapsulation efficiency of 36.63% and a 48 h-sustained release in vitro. Superior bio-compatibility was confirmed with red blood cells (RBCs) and HEK 293 cells. Due to the RBCM coating, RPTR-701Ns on one hand significantly reduced phagocytosis by RAW 264.7 cells as compared to PTR-701Ns, showing an immune escape effect. On the other hand, RPTR-701Ns had an advanced exotoxins neutralization ability, which helped reduce the damage of MRSA exotoxins to RBCs by 17.13%. Furthermore, excellent in vivo bacteria elimination and promoted wound healing were observed of RPTR-701Ns with a MRSA-infected mice model without causing toxicity. In summary, the novel delivery system provides a synergistic antibacterial treatment of both sustained release and bacterial toxins absorption, facilitating the incorporation of TR-701 into modern nanotechnology.

## 1. Introduction

Multiple drug resistance (MDR) of bacteria has posed significant clinical challenges to human health worldwide [1,2]. Methicillin-resistant staphylococcus aureus (MRSA) possesses high toxicity and causes serious diseases such as purulent skin and soft tissue infections, pneumonia, bacteremia, and osteomyelitis, and represents the biggest therapeutic hurdles [3]. In recent years, the increased incidence of MRSA-infections has been exploited in hospitals, nursing homes, medical institutions and communities [4]. The treatment of MRSA has been a higher demand. Among all therapies, vancomycin is the first choice for the treatment of serious gram-positive infections. However, with the emergence of vancomycin-resistant strains, its efficacy has been restricted [5]. Clearly, there is an urgent requirement for novel and effective therapeutic strategies for MRSA treatment.

Tedizolid phosphate (TR-701) is a novel antibiotic approved for acute bacterial skin and skin structure infection (ABSSSI) caused by gram-positive bacteria, including methicillin-sensitive staphylococcus aureus (MSSA) and MRSA [6,7]. It has demonstrated high efficacy and safety both in vitro and in vivo [8,9,10,11]. A large number of clinical trials have proved TR-701 possessing good antibacterial efficacy [12,13]. Nevertheless, few studies have been performed on the novel dosage form of TR-701 in recent years. It was only loaded into cationic liposomes against bacterial pneumonia, but there was a lack of pharmacodynamic studies to verify the in vivo efficacy [14]. There is actually no research on the new dosage form of TR-701 for antibacterial infection therapy.

For better treatment of bacterial infection, numerous delivery systems have been explored and nanomedicine becomes a promising delivery strategy of antimicrobial [15,16]. Metal nanoparticles such as silver, titanium dioxide and gold showed good antibacterial efficacy [17,18], but the risk of leaking out metallics may cause damage to health [19]. For higher bio-compatibility, polymeric nanoparticles are preferred. Poly (lactic-co-glycolic acid) (PLGA), due to its high biocompatibility, is a most commonly used polymer material for nanoparticles [20,21]. For longer circulation and less immune recognition of nanoparticle drugs, diverse strategies are reported, such as modifying the surface with polyethyleneglycol (PEG) or cell-penetrating peptides [22]. Among them, red blood cell membrane (RBCM), as a natural bio-membrane extracted from red blood cells (RBCs), attracted extensive attention as a drug carrier [23,24,25]. RBCM is expected to help achieve a long circulation of drugs because of the presence of special proteins like CD47 contributing to lower uptake by macrophages [26,27]. Liangfang Zhang et al. [28] demonstrated that RBCM helped reduce macrophage uptake of PLGA nanoparticles with longer half-life than PEG-modified nanoparticles. On the other hand, RBCM-based nano-strategies also provide an attractive platform for the anti-infection application. It is reported to be lysed by various bacterial exotoxins and promote interaction of drug loaded nanoparticle and pathogens [29]. “Nano-sponge” with polymeric core and outer RBCM was reported to detoxify bacterial toxins, including melittin, MRSA α-hemolysin (Hlα), hemolytic listeriolysin O (LLO), group A streptococcus streptolysin O (GAS) and group B streptococcus β-hemolysin/cytolysin (GBS) etc [30,31,32,33]. For drug-loaded RBCM-nanoparticles, a super-molecular gelatin-coated vancomycin nanoparticle system modified with RBCM was developed for adaptive and “on-demand” antibiotic delivery [34]. RBCM imparted its biomimetic properties with immune escaping function, as well as toxins absorption. Upon the rapture of RBCM, gelatin specifically degraded and released the drug at the inflammatory site to achieve an antibacterial effect. However, there was a lack of in vivo verification and it was doubtful whether the delivery system would make sense in infected models. Yue Zhang [35] et al. synthesized a redox-responsive crosslinker to load vancomycin, and then coated it with RBCM. The nano-system cleared MRSA-associated toxins extracellular and had an accelerated drug release profile in intracellular reducing environment for effective bacterial inhibition. However, the synthesis of the nanomaterial was quite complicated and the safety of the carrier was unknown. Ange Lin et al. [36] designed RBCM coated gelatin nanoparticles loaded Ru−Se (Ru−Se @GNP-RBCM), in which each part of the bacteria-responsive nano-system possessed a synergistic effect in bacterial infection treatment. This research was relatively complete, but the preparation of Ru-Se was time-consuming and relevant characterization was needed to determine the success of the synthesis. Meanwhile, the antibacterial effect was no better than vancomycin, which has faced serious problem of drug resistance. In summary, abundant studies have proved that RBCM-based nanoparticle systems are playing a promising role in delivering antibiotics [37]. However, there is still a lot of improvements to make, such as exploring dosage form of novel antibiotics, relatively simple preparation, safety and efficacy studies both in vitro and in vivo and so on.

In this study, TR-701 was encapsulated into PLGA nanoparticles and wrapped with RBCM to form RBCM coated-TR-701 loaded PLGA nanoparticles (RPTR-701Ns) according to Figure 1. The RPTR-701Ns were characterized and evaluated for their antibacterial efficacy and safety both in vitro and in vivo. The proposed delivery system was obtained through a relatively simple method, and it has been verified both in vivo and in vitro to possess high bio-compatibility and antibacterial efficacy with toxins adsorption capability.

## 2. Materials and Methods

### 2.1. Materials

TR-701 was purchased from Shanghai Macklin Biotechnology Co., Ltd. OH–PLGA–COOH 50/50 (PLGA, Mw = 20,000) was purchased from Jinan Daigang Biomaterial Co., Ltd. Poloxamer188 (P188) was purchased from BASF. DiO (green fluorescent probe), Cell counting kit-8 (CCK-8), 4% paraformaldehyde, antifade mounting medium, PBS, 10 × PBS were purchased from Biyotime Biotechnology Co., Ltd. Saline and agar powder were purchased from Shanghai Yuanye Biotechnology Co., Ltd. 2% Phosphotungstic acid was purchased from Beijing Solarbio Science and Technology Co., Ltd. Carbon support film (230 mesh) was purchased from Beijing Zhongjingkeyi Technology Co., Ltd. Dialysis bag (36 mm, MD44, MW:3500) was purchased from Shanghai labeen Biotechnology Co., Ltd. 0.25% Trypsin-EDTA, penicillin-streptomycin and fetal bovine serum (FBS) were purchased from Gibco, USA. DMEM high glucose medium was purchased from HyClone Biochemical Products (Beijing) Co., Ltd. Dimethyl sulfoxide (DMSO) was purchased from Shanghai MP Biomedical Co., Ltd. Sodium chloride (>99.5%) was purchased from Shanghai Sangon Biotechnology Co., Ltd. Sodium bicarbonate (>99%) was purchased from Shanghai Saen Chemical Technology Co., Ltd. Yeast extract and tryptone were purchased from Oxoid, USA. Acetone and EDTA-2Na were purchased from Sinopharm Chemical Reagent Co., Ltd.

### 2.2. Cells, Bacteria and Animals

Murine RAW 264.7 and HEK 293 cells were obtained from the Shanghai Cell Bank of the Chinese Academy of Sciences. RAW 264.7 cells were cultured in DMEM medium containing 10% FBS at 37 °C and 5% CO_2_. HEK 293 cells were cultured in DMEM medium containing 10% FBS and 1% penicillin-streptomycin at 37 °C and 5% CO_2_.

MRSA USA 300, MRSA ATCC 43300 and MSSA NTCT 8325 were purchased from American Type Culture Collection (ATCC) and cultured in LB medium at 37 °C. LB (Luria-Bertani) medium is a commonly used medium for culturing bacteria. It was prepared by dissolving 10 g tryptone, 5 g yeast extract, and 10 g sodium chloride together in 1 L of pure water and then got sterilized with an autoclave.

SD rats and BALB/c mice were purchased from Shanghai Jiesijie Experimental Animal Co., Ltd. All animal experiments were performed in accordance with the Guidelines for Care and Use of Laboratory Animals of Shanghai Jiao Tong University and approved by the Animal Ethics Committee (Number: 202004020 Date: 10 April 2020).

### 2.3. Preparation of PTR-701Ns

PTR-701Ns were prepared through an emulsion solvent evaporation method [38]. 15 mg PLGA was dissolved into 1 mL acetone as the organic phase. A sum of 150 mg NaHCO_3_ was dissolved in 5 mL ultrapure water to obtain a 30 mg/mL NaHCO_3_ solution, which was used as the solvent for TR-701. Then, 30 mg TR-701 was dissolved in 1 mL of the above NaHCO_3_ solution to obtain a 30 mg/mL TR-701 solution. For aqueous phase, 1 mL of the TR-701 solution was added into 9 mL PBS pre-dissolved with P188, with final concentration of TR-701 and P188 as 3 mg/mL, and 10 mg/mL, respectively. Then, the organic phase was slowly added into the aqueous phase under the condition of magnetic stirring at 420 rpm for 3 h at room temperature to remove acetone and obtain PTR-701Ns.

### 2.4. Preparation of RBCM

RBCM was obtained by low-osmotic hemolysis method. Firstly, blood was obtained from SD rats through abdominal aorta and 500 μL blood was added into each tube. The blood was extracted by centrifugation (4800 rpm, 4 min, 4 °C) to obtain RBCs. A total of 1 mL PBS was added and washed the cells twice to make sure the RBCs were clear. To collect RBCMs, 900 μL of 0.2 mM EDTA-2Na solution was added to split the RBCs, followed by centrifugation (13,200 rpm, 10 min, 4 °C) to remove hemoglobin. The above steps were repeated for 5–10 times until the supernatant was colorless and clear. Finally, RBCM was dispersed in EDTA-2Na solution and stored at −80 °C.

### 2.5. Preparation of RPTR-701Ns

RPTR-701Ns were prepared with an ultrasound method. For this purpose, 500 μL of RBCMs solution (dispersed in PBS) was added into 10 mL of PTR-701Ns solution under ultrasound at 100 waves for 3 min.

### 2.6. Characterization and Stability Test

The particle size and polydispersity index (PDI) of PTR-701Ns and RPTR-701Ns were determined by Malvern Zetasizer (ko1⁄4633 nm, He-Ne, 4.0 Mw, Malvern instruments Ltd., Malvern, UK). To test the stability of the nanoparticles, the particle size and PDI of PTR-701Ns and RPTR-701Ns were measured daily for 7 days at both room temperature and 4 °C.

### 2.7. Morphological Observation

The morphology of PTR-701Ns and RPTR-701Ns were performed by transmission electron microscopy (TEM, Thermo Fisher Scientific, Shanghai, China). 10 μL PTR-701Ns and RPTR-701Ns solutions were added onto carbon-supported coppers. After 10 min incubation, they were removed from the edges of the copper with filter paper. Then, 10 μL of 2% phosphotungstic acid was dropped onto the coppers to dry the samples. After 30 s, the excess stain was removed. The morphology of the air-dried samples was observed by TEM.

### 2.8. Drug Loading Capacity Test

Encapsulation efficiency (EE) and drug loading (DL) were measured to determine the drug loading capability of the nanoparticles.

3 mL PTR-701Ns solution was added into a dialysis bag, and then immersed in 50 mL PBS to remove unloaded drugs with magnetic stirring at 100 rpm at 37 °C for 3 h. Then, 1 mL dialysate was collected and filtered with a 0.22 μm microporous membrane. PLGA nanoparticles without loading drugs were also prepared and dialyzed to obtain blank dialysate. 250 μg/mL TR-701 dissolved in blank dialysate was used as the standard. All samples were injected into the HPLC for analysis. The HPLC analysis system was established by Agilent 1200 high performance liquid chromatography with Agilent Ultimate^®^ XB-C18 column (5 μm, 4.6 × 250 nm). As C18 column is a reversed phase column, substances with large polarity would flow down from the column and get separated and measured, according to the principle of similar compatibility. The effluent was detected at 299 nm with the temperature of 35 °C. A mixed solution of 25 mM ammonium acetate (A) and acetonitrile tetrahydrofuran 9:1(*v/v*) (B) was used as a mobile phase, and the flow rate was 1 mL/min. The gradient elution program was set as following: 0–1.0 min, 10%B; 1–25.0 min, 10%–50%B; 25.0–35.0 min, 50%–60%B; 35.0–40.0 min, 60%–70%B; 40.0–40.1 min, 70%–10%B; 40.1–45.0 min, 10% B.

EE and DL were calculated by the following formula:(1)$$\mathrm{EE}\;\left(\%\right)=\frac{(M1\;-\;M2)}{M1}\times 100\%$$
(2)$$\mathrm{DL}\;\left(\%\right)=\;\frac{\left(M1-M2\right)}{M3}\times 100\%$$

M1: the amount total TR-701 used; M2: the amount unloaded TR-701 dialyzed out; M3: the amount of the PTR-701Ns.

### 2.9. In Vitro Release Study

To test the in vitro release, 3 mL TR-701, PTR-701Ns and RPTR-701Ns solutions were added into dialysis bags, and then put into 50 mL PBS to release drug under 100 rpm magnetic stirring and 37 °C. A total of 1 mL of dialysate was taken out at the time point of 0.5, 1, 2, 4, 8, 12, 36, 48 h with another 1 mL PBS replacement. Then, the dialysate was injected into HPLC to determine the release ratio. This method was same as mentioned in Section 2.8 for HPLC analysis.

### 2.10. Hemolysis Test

RBCs were collected from SD rats and 2% (*v*/*v*) RBCs suspension was made by mixing RBCs with normal saline. In vitro, RBCs usually maintain in good condition in normal saline as it provides proper osmotic pressure. RBCs will be destroyed if the osmotic pressure is too high or too low. Different amounts of RBCs suspension, normal saline, ultrapure water and RPTR-701Ns solution were added into tubes according to Table 1. In Tube 6, normal saline mixed with 2% RBCs was used as the negative control group, where all RBCs would not get hemolyzed. In Tube 7, ultrapure water was mixed with 2% RBCs to make all RBCs get damaged as the positive control group. After incubation for 3 h and 24 h at 37 °C, the absorbance of the supernatant was tested with ultraviolet spectrophotometry at 540 nm to determine the hemolysis of RBCs in each group. By comparing the degree of hemolysis of the RBCs treated with different concentrations of RPTR-701Ns and the control groups, damage on RBCs of RPTR-701Ns could be found. The hemolysis ratio (HR) was calculated as follows:(3)$$\mathrm{HR}\;(\%)=\frac{A{\left(540\right)}_{\mathrm{Sample}}\;-\;A{\left(540\right)}_{\mathrm{Tube}\;6}}{A{\left(540\right)}_{\mathrm{Tube}\;7}\;-\;A{\left(540\right)}_{\mathrm{Tube}\;6}}\;\times \;100\%$$

### 2.11. In Vitro Cytotoxicity Assay

To evaluate the bio-compatibility of nanoparticle materials (PLGA and RBCM) and drug loading nanoparticles, cytotoxicity test was carried out by CCK-8. Human embryonic kidney 293 (HEK 293) cells were employed as normal cell lines in this section. The HEK 293 cells were obtained from Shanghai Cell Bank of the Chinese Academy of Sciences.

For the cytotoxicity of nanoparticle materials, PLGA nanoparticles (PNs) and RBCM coated PLGA nanoparticles (RPNs) were prepared without TR-701, and diluted with DMEM complete medium with 0.5, 2.5, 5, 10, 20, 30, 40 μg/mL PLGA concentration. For drug loading nanoparticles, TR-701, PTR-701Ns and RPTR-701Ns were diluted with DMEM complete medium with TR-701 concentration of 1, 5,10, 20, 30, 40, 80 μg/mL. HEK 293 cells (approximately 2 × 10^5^ cells/mL, 100 μL) were seeded in 96-well plates for 24 h at 37 °C. The DMEM medium was then removed and 100 μL of the samples were added and incubated with the cells for another 24 h incubation. Afterward, 100 μL of CCK-8 solution (1:9 *v*/*v* diluted with DMEM complete medium) was added. After 2 h incubation, the plates were tested for optical density (OD) at 450 nm (600 nm for reference) through a microplate reader (Diken Trading Co. Ltd., Shanghai, China). Cells cultured with only DMEM complete medium was set as a control group. DMEM complete medium was set as a blank group. The cell survival ratio was calculated according to the following formula:(4)$$\mathrm{Cell}\;\mathrm{survival}\;\mathrm{ratio}\;(\%)=\frac{{\mathrm{OD}}_{\;\mathrm{sample}}\;-{\;\mathrm{OD}}_{\;\mathrm{blank}}}{{\mathrm{OD}\;}_{\mathrm{control}\;}-{\;\mathrm{OD}}_{\;\mathrm{blank}}}\;\times \;100\%$$

### 2.12. Macrophage Uptake Study

The immune-escaping ability of RPTR-701Ns was tested in by murine RAW 264.7 cells. PLGA was labeled by DiO dye. And PTR-701Ns and RPTR-701Ns were prepared by DiO-labelled PLGA. Raw 264.7 cells (2 × 10^5^ cells/mL, 1 mL) were seeded in a 24-well plates in DMEM complete medium for 24 h at 37 °C. Then, the DMEM medium was removed and replaced by 1 mL of serum-free DMEM medium. After starving the cells for 1 h, the medium was removed and 1 mL of the DiO-labelled PTR-701Ns and RPTR-701Ns (30 μg/mL of PLGA) were added and incubated for 2 h. For qualitative observation, the cells were washed with PBS for 3 times and fixed by 4% paraformaldehyde for 15 min. After being washed for another 3 times, the cells were treated by an anti-fluorescence quenching mounting solution and then observed by laser scanning confocal microscope (LSCM, Leica Microsystems Ltd., Solms, Germany). For quantitative detection, the cells were digested by trypsin and then collected. After being washed with PBS for 3 times, the cells were resuspended in 500 μL PBS and detected by a flow cytometer (Beckman Coulter Inc., Brea, USA) at the FITC channel.

### 2.13. In Vitro Antibacterial Efficacy Study

MRSA USA 300, MRSA ATCC 43300 and MSSA NTCT 8325 were chosen to evaluate the in vitro anti-bacterial efficacy of RPTR-701Ns in short stage. The three kinds of bacteria were cultured in fresh LB medium for 18 h at 37 °C on a shaker at 200 rpm. Then the concentration of bacterial suspension was measured and diluted, and 100 µL (about 1×10^6^ CFU/mL) bacterial suspension was added into a 96-well plate. Afterwards, 100 µL TR-701, PTR-701Ns, RPTR-701Ns with TR-701 concentration of 16, 8, 4, 2, 1, 0.5, 0.25, 0.125 μg/mL were added and incubated for 18 h at 37 °C. OD 600 was measured by a microplate reader. 100 µL of LB medium mixed with 100 µL of bacteria suspension was used as the positive control and 200 µL of LB medium was used as the negative control. The bacteria viability ratio was calculated as follows:(5)$$\mathrm{Bacteria}\;\mathrm{viability}\;\mathrm{ratio}\;(\%)=\frac{{\mathrm{OD}\;}_{\mathrm{sample}}-{\mathrm{OD}}_{\;\mathrm{negative}}}{{\mathrm{OD}\;}_{\mathrm{positive}}-{\mathrm{OD}}_{\;\mathrm{negative}}}\;\times \;100\%$$

### 2.14. Bacterial Exotoxins Removing Capability Study

The exotoxins removing capability of RPTR-701Ns was carried out by measuring the anti-hemolysis activity of RBCs exposed to the supernatant of bacterial medium mixed with or without RPTR-701Ns. Firstly, MRSA USA 300 (as model bacteria) was cultured overnight and adjusted into OD 600 = 0.5. The supernatant was obtained by centrifugation (5000 rpm,10 min, 4 °C) and filtered with a 0.22 μm membrane to remove bacteria. Then, 200 μL TR-701, PTR-701Ns, RPTR-701Ns (TR-701 concentration 150 μg/mL) were added into 500 μL of the supernatant respectively and incubated for 2 h at 37 °C to detoxify exotoxins. Afterwards, 25 μL of whole blood was added and incubated for another 2 h. Finally, the supernatant was obtained by centrifugation (4800 rpm, 4 min, 4 °C) and OD 540 was measured through a microplate reader to quantify the released hemoglobin. LB medium was used as the negative control, and the 1% (*v*/*v*) Trixton-100 LB medium was the used as the positive control. The hemolysis ratio (HR) was calculated according to Equation (3).

### 2.15. In Vivo Anti-Infection Study

A wound infection model was established with Balb/C mice. A total of 24 healthy male Balb/C mice (about 20 g) were fed regularly for a week and divided into 4 groups (6 per group): Control group, TR-701 group, PTR-701Ns group and RPTR-701Ns group. After removing the hair on the back, a round wound was created with surgical scissors on each mouse. Then, bacteria suspension (MRSA USA 300, 50 μL, 10^9^ CFU/mL) was spread onto the wound. 24 h after wound development, mice were given TR-701, PTR-701Ns and RPTR-701Ns by caudal vein every day for 7 days, at a dose of 26 μg/g. Mice in the control group were given the same amount of PBS. The administration was stopped after 7 days, and then all mice continued to be observed and recorded in the following 4 days without treatment. Each day, bodyweight was recorded, and the wound area was photographed to evaluate the wound healing. On day 11, blood samples of all mice were collected and mice were sacrificed to collect wound tissues and main organs.

#### 2.15.1. In Vivo Anti-Bacterial Study

A daily picture of the wound area of each mouse was photographed to calculate the area of the wound to evaluate healing rate. Moreover, infected skin samples were collected and ground, then spread on LB agar plates to culture at 37 °C for 18 h. On the next day, the colony-forming units were counted. Also, the wound skin on the last day was also H&E (Hematoxylin Eosin) stained for histological analysis.

#### 2.15.2. In Vivo Safety Study

For in vivo safety evaluation, the daily body weight of the mice was recorded. Organs including heart, liver, spleen, lung and kidneys were collected and weighted. Organ coefficient was calculated as weight of organ/body weight. Blood of all mice was collected to evaluate for RBCs, white blood cells (WBCs), Hematocrit (HCT) and Hemoglobin (HGB) index. The main organs were stained for histological analysis.

### 2.16. Statistics and Data Analysis

All data were presented as mean ± standard deviation (SD). Significant differences were analyzed by *t*-test of GraphPad Prism Software (Version 8.2.1, GraphPad Software Inc., San Diego, USA). * indicates a significant difference. * means *p* < 0.05, ** means *p* < 0.01, *** means *p* < 0.001, **** means *p* < 0.0001. The area of wound was calculated by Image J 1.8.0 Software.

## 3. Results

### 3.1. Preparation and Stability of PTR-701Ns and RPTR-701Ns

The diameters of PTR-701Ns and RPTR-701Ns were 177.90 ± 1.61 nm and 192.50 ± 5.85 nm, respectively (Figure 2A). It was worth noticing that the increment in diameter was approximately equals to the thickness of RBCM, that was about 7 nm [39]. It indicated that RBCM was successfully coated onto the surface of PTR-701Ns. PDI of PTR-701Ns and RPTR-701Ns was 0.0084 ± 0.011 and 0.13 ± 0.021 (Figure 2B), showing both PTR-701Ns and RPTR-701Ns were homogeneous and dispersed well.

As to stability test of the nanoparticles, the changes of size and PDI have been recorded within 7 days at room temperature and 4 °C. At room temperature, the size of PTR-701Ns and RPTR-701Ns fluctuated slightly between 172 to 182 nm and 185 to 210 nm (Figure 2C), and the PDI of all samples maintained below 0.2 (Figure 2D). However, for 4 °C, the particle size of both PTR-701Ns and RPTR-701Ns increased significantly within 4 days and the flocculation occurred which may be caused by the drug precipitation. Therefore, it suggested that PTR-701Ns and RPTR-701Ns were stable in size and PDI for 7 days at room temperature.

### 3.2. Morphological Observation of PTR-701Ns and RPTR-701Ns

To further confirm the formation of the nanoparticles, PTR-701Ns and RPTR-701Ns were observed by TEM (Figure 3). PTR-701Ns were spherical in shape with a smooth surface and distributed uniformly. As the RBCM wrapped around PTR-701Ns, RPTR-701Ns presented a core-shell spherical structure with relatively blurred edges. Moreover, the particle size of PTR-701Ns and RTR-701Ns measured by TEM was about 100–150 nm, which was slightly smaller than the one measured by Malvern particle sizer. Such difference may occur due to the removal of the water film on the outside around the nanoparticles after the samples were dried.

### 3.3. Drug Loading Capacity and In Vitro Release Study

As important indicators, EE and DL were examined to evaluate the drug loading capability of the nanoparticles. Analyzed by HPLC, the nanoparticles showed good EE and DL of 36.63%, and 24.42% on average, respectively. In vitro release was examined to evaluate whether the nanoparticles could achieve sustained release (Figure 4). All three groups released the drug faster in the first 4 h. The TR-701 group achieved 97.28% of release in 8 h with full release in 12 h. For the nanoparticles, the pace slowed down significantly after 4 h of relative fast release, which may be caused by free TR-701. The cumulative release of PTR-701Ns was 91.48% at 12 h, and the residue continued to release 96.75% at 48 h. The released rate of RPTR-701Ns slowed down to 85.11% at 12 h and achieved 93.37% in 48 h. In summary, the release pace of RPTR-701Ns at different time point was slower as compared to PTR-701Ns and TR-701. Such sustained release of RPTR-701Ns may give thanks to RBCM, which formed a physical barrier around TR-701 to further prevent the rapid penetration of drugs.

### 3.4. Hemolysis Test

Since RPTR-701Ns were administrated intravenously, it is important for RPTR-701Ns to show good bio-compatibility with RBCs. In Figure 5, Tube 7 was set as a positive control group where all RBCs were ruptured, and Tube 6 was a negative control group where RBCs were not lysed. After incubated with different concentrations of RPTR-701Ns for 3 h, and 24 h respectively, all RBCs sample group were in good condition, showing no prominent hemolysis. Further, the absorbance of the supernatant of the sample groups was measured by an ultraviolet spectrophotometer. The hemolysis ratio of RPTR-701Ns of all different concentrations was lower than 1.5%, which indicates that RPTR-701Ns do not cause cell hemolysis in short term or long term after administration and are safe for intravenous injection.

### 3.5. In Vitro Cytotoxicity Assay

Cytotoxicity of nanoparticle material and TR-701 loaded nanoparticles were carried out by CCK-8 method. Normal cell lines (HEK 293 cells) was used to check whether the nanoparticles would have toxicity to normal cells. PNs and RPNs were nanoparticle systems prepared without TR-701 loaded. As shown in Figure 6A, the viability of HEK 293 cells incubated with PTNs and RPTNs were higher than 95% at PLGA concentration of 0.5–40 μg/mL. It indicated that the both PLGA and RBCM had negligible cytotoxicity and high compatibility. For drug loading nanoparticles, the viability of cell was roughly dose-dependent (Figure 6B). At low concentrations of TR-701 (1–20 μg/mL), the cell viability of each group reached 100%, indicating that neither drugs nor the nanoparticles would cause damage to cells. Whereas, with the higher TR-701 concentration, the survival of cells declined slightly. It is speculated that the toxicity was caused by the drug. After the drug was coated into RPNs, the toxicity did not increase but slightly reduced, which predicted the high biocompatibility of RPTR-701Ns.

### 3.6. Macrophage Uptake Study

After being coated with RBCM, it is necessary to make sure whether RPTR-701Ns possess the biomimetic property of RBCM with high immune-compatibility [40]. Capability of immune-escape helps drug loaded nanoparticles to circulate longer, and thus, deliver more drug at targeted site. A macrophage uptake study was carried out by anti-phagocytosis against Raw 264.7 cells. In order to track the nanoparticles, PLGA was labelled with DiO (a lipophilic membrane dye). The fluorescence intensity of the RPTR-701Ns group was significantly reduced in comparison with the PTR-701Ns group by LSCM (Figure 7A). By quantitation, Figure 7B indicated smaller offset of fluorescence of RPTR-701Ns (red) than PTR-701Ns (blue). The mean fluorescence intensity of PTR-701Ns and RPTR-701Ns groups were 87.1 and 78.53, and the median fluorescence intensity were 70.53 and 62.43, respectively (Figure 7C,D), showing a significant decrease of uptake of RPTR-701Ns. Both results of LSCM and the flow cytometer proved that after being coated with RBCM, RPTR-701Ns could help nanoparticles avoid macrophage uptake, which may be related to the existence of particular proteins on RBCM [41].

### 3.7. In Vitro Antibacterial Efficacy Study

As TR-701 is a kind of antibiotic, it is necessary to evaluate the in vitro antibacterial effect of the formulations. Three different species of bacteria were used, MSSA NTCT 8325 was a species of methicillin-sensitive staphylococcus aureus, MRSA USA300 and MRSA 43300 were methicillin-resistant staphylococcus aureus. It can be seen from Figure 8 that, for different staphylococcus aureus species, both RPTR-701Ns and PTR-701Ns had an antibacterial efficacy equivalent to TR-701. This indicated that RPTR-701Ns could release drug well even with the coating of PLGA and RBCM.

For MRSA USA 300, MRSA 43300 and MSSA 8325, the minimum inhibitory concentration (MIC) in 18 h of the three preparations were 4 μg/mL, 4 μg/mL and 2 μg/mL, respectively. The preparations showed high bacteria killing effect on staphylococcus aureus at low a dosage. The results proved that the RPTR-701Ns could achieve good antibacterial inhibition in short stage.

### 3.8. Bacterial Exotoxins Removing Capability Study

Exotoxins secreted by bacteria can cause various diseases. RBCM is reported to absorb such exotoxins, and help drug-loaded nanoparticles get exposed to bacteria and promote antibacterial efficacy [30]. Exotoxins-removing capability of RPTR-701Ns was examined through anti-hemolytic experiments by measuring hemoglobin released from damaged RBCs by exotoxins. MRSA USA 300 was used as a model of bacteria. The result showed that the hemolysis ratio of RBCs decreased remarkably from 82.94% to 65.81% after the administration of RPTR-701Ns (Figure 9A,C). The group treated with TR-701 and PTR-701Ns presented a high hemolysis ratio, similar to the one without treatment. Besides, the color of RBCs in the control group was black in comparison with the red ones in RPTR-701Ns group, which were similar to natural RBCs (Figure 9B), indicating the excellent ability of adsorbing bacterial toxins of RPTR-701Ns, thereby, protecting the morphology and bio-activity of RBCs. The above experiments implied that RPTR-701Ns possessed exotoxins removal capacity due to the RBCM.

### 3.9. In Vivo Ant-Infection Study

To evaluate in vivo anti-bacterial efficacy, MRSA USA 300 wound infection model was established. We demonstrated TR-701, PTR-701Ns and RPTR0701Ns with same amount of drug at same dosing interval to investigate difference on therapeutic effects of different dosage forms. After 11 days of treatment, a better wound healing rate in the group treated with RPTR-701Ns was showed (Figure 10B,C). At the end of the treatment period, the wound areas of RPTR-701Ns group was just about 22.65% of the initial wound area, which was dramatically lower than the control and TR-701 group which were 44.01%, and 32.13%, respectively. The facilitated wound healing indicated a better therapeutic effect of TR-701Ns.

Furthermore, the skin of the wound was collected and then spread on LB plates to quantify the member of the bacteria colony remaining in the wound. As shown in Figure 10D,E, the colonies grown on the plates of the RPTR-701Ns group were significantly less than the rest groups. The RPTR-701Ns group significantly inhibited MRSA growth and showed a synergistic antibacterial activity of both RBCM and the drug. On one hand, RBCM helped RPTR-701Ns to achieve the effect of absorption of exotoxins, which was also verified by in vitro experiments. On the other hand, RPTR-701Ns have a sustained release of TR-701 compared to the naked drug. It was speculated that PTR-701Ns were cleared by macrophages in vivo. But with the coating of RBCM, RPTR-701Ns maintained a longer circulation in vivo, so as to promote the aggregation of the drug at infected site and achieve a better therapeutic effect [36].

In tissue analysis of wound (Figure 10F), compared to the skin of the control group which showed extensive epidermal necrosis, capillary arrangement disorders (black arrow), and massive neutrophil granulocyte infiltration (yellow arrows), the three treatment groups presented healing conditions. After treated with RPTR-701Ns, mature granulation tissue and thick newborn epidermis (blue arrows) appeared. All the results indicated a significant potential of RPTR-701Ns to MRSA-infected wound treatment.

### 3.10. In Vivo Safety Study

In addition to advantageous antibacterial efficacy, RPTR-701Ns should have high in vivo safety. During the whole treatment, the body weight of the mice in each group increased slowly over time (Figure 11A), and there was no significant difference between the treated groups and the control group. The organ coefficients were normal without significant difference (Figure 11B). RBCs, WBCs, HCT and HGB index were all within the reference range (Figure 11C–F). Moreover, histological analysis was conducted by H&E staining (Figure 11G). Main organs (heart, liver, spleen, lung and kidneys) of treated groups showed normal as the control group. Overall, administration of drug for 7 continuously days showed no apparent systemic toxicity. Therefore, RPTR-701Ns are promising reagents for antibiotic therapy with high biocompatibility.

## 4. Conclusions

Previous studies on RBCM-based delivery systems for antibacterial treatments have achieved synergistic antibacterial and responsive release effects, compared with the current used treatments. For example, once vancomycin was loaded into a redox-responsive crosslinker coated with RBCM, it showed better intracellular bacterial inhibition than the ‘naked’ vancomycin. The RBCM provided the preparation with toxins clearance and immune escaping capability [35]. For MRSA infected wound, a RBCM-coated Fe_3_O_4_ nanoparticles (RBC@Fe_3_O_4_) had superior treatment efficacy than the Fe_3_O_4_ itself [42]. The RBC@Fe_3_O_4_ acted as nano-sponges to trap bacterial toxins and then kill them all with a photothermal effect of Fe_3_O_4_. Furthermore, a Ru−Se @GNP-RBCM delivery system provided a synergistic effect with responsive release of the antibacterial agents for MRSA infective therapy [36]. Better MRSA inhibition and wound healing rated was proved after administration of Ru−Se @GNP-RBCM than Ru-Se @GNP. These results proved that the RBCM-based delivery system had more efficient treatment for MRSA infections. However, there were still a lot of improvements to make, such as exploring dosage form of novel antibiotics, relatively simple preparation, safety and efficacy studies both in vitro and in vivo, etc.

However, there was a lack of simply made biomimetic delivery system for MRSA infections with high safety and antibacterial efficacy both in vivo and in vitro. As a novel drug against MRSA, TR-701 has been reported to be effective clinically.

Herein, a biomimetic delivery system was developed to provide TR-701 with a new dosage form and endow it with a sustained release and better antibacterial therapy. PTR-701Ns were prepared by loading TR-701 into PLGA nanoparticles, and then RPTR-701Ns were developed by coating natural RBCM on the surface of the spherical-shaped PTR-701Ns, that helped to reduce the recognition of macrophages. The core-shell structure RPTR-701Ns remained stable in seven days at room temperature, with an average EE and DL of 36.36% and 24.40%, respectively. A sustained release was observed for 48 h of RPTR-701Ns as compared with the quick release of TR-701. A high compatibility of RPTR-701Ns was shown with RBCs and HEK 293 cells through hemolysis and CCK-8 test. In antibacterial efficacy, RPTR-701Ns presented similar inhibition with TR-701 to both MRSA and MSSA in short stage with significant exotoxins absorbing capability. Furthermore, in vivo study showed RPTR-701Ns promoted healing rate of MRSA-infected wound than TR-701 and can better inhibit the growth of MRSA in skin wound. The treatment of RPTR-701 was also proved safe to blood and organs.

Due to the proposed advantages, including its relatively simple method, high bio-compatibility and better antibacterial efficacy, both in vivo and in vitro, RPTR-701Ns were promising in MRSA-infection treatments. In the future, RPTR-701Ns as a novel biomimetic dosage form of TR-701, is expected to be a bio-therapy in drug-resistant bacterial infections with high potentials.

## Figures and Tables

**Figure 1 pharmaceutics-13-00099-f001:**
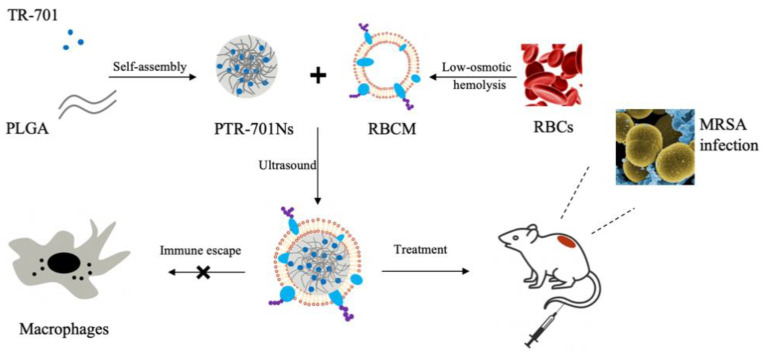
Schematic illustration of preparation and application of RPTR-701Ns.

**Figure 2 pharmaceutics-13-00099-f002:**
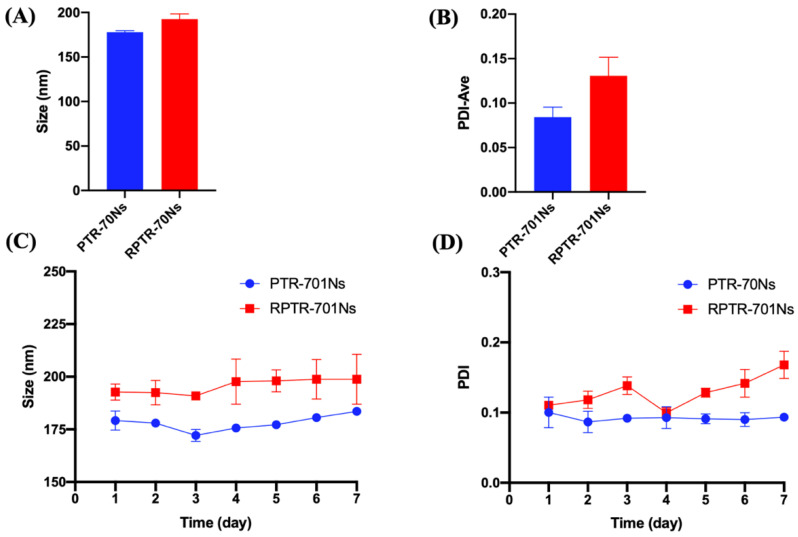
Characterization of nanoparticles. (**A**) Particle size, (**B**) PDI, **(C**) particle size and (**D**) PDI change in 7 days at room temperature of PTR-701Ns and RPTR-701Ns. Data were presented as the mean ± SD (*n* = 3).

**Figure 3 pharmaceutics-13-00099-f003:**
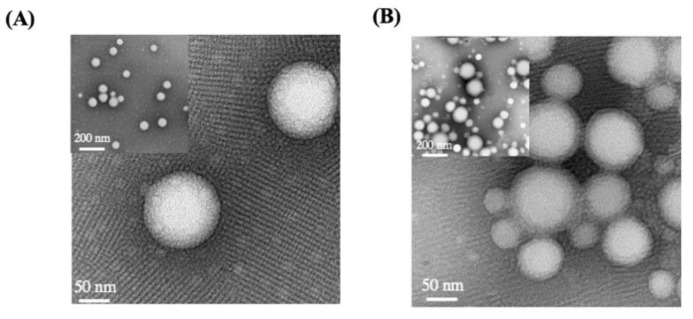
TEM images of; (**A**) PTR-701Ns and (**B**) RPTR-701Ns.

**Figure 4 pharmaceutics-13-00099-f004:**
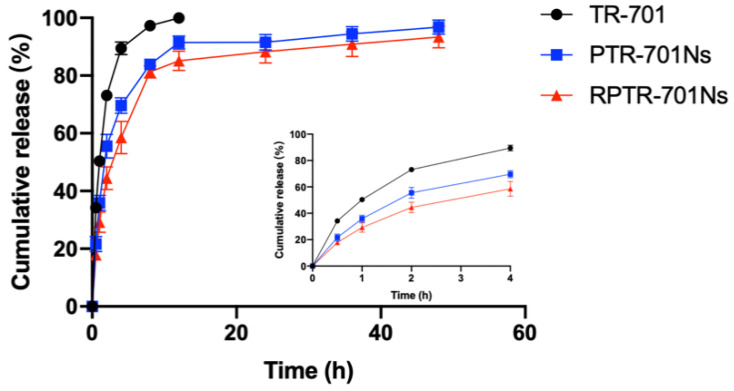
In vitro release of TR-701, PTR-701Ns and RPTR-701Ns. Data were presented as the mean ± SD (*n* = 3).

**Figure 5 pharmaceutics-13-00099-f005:**
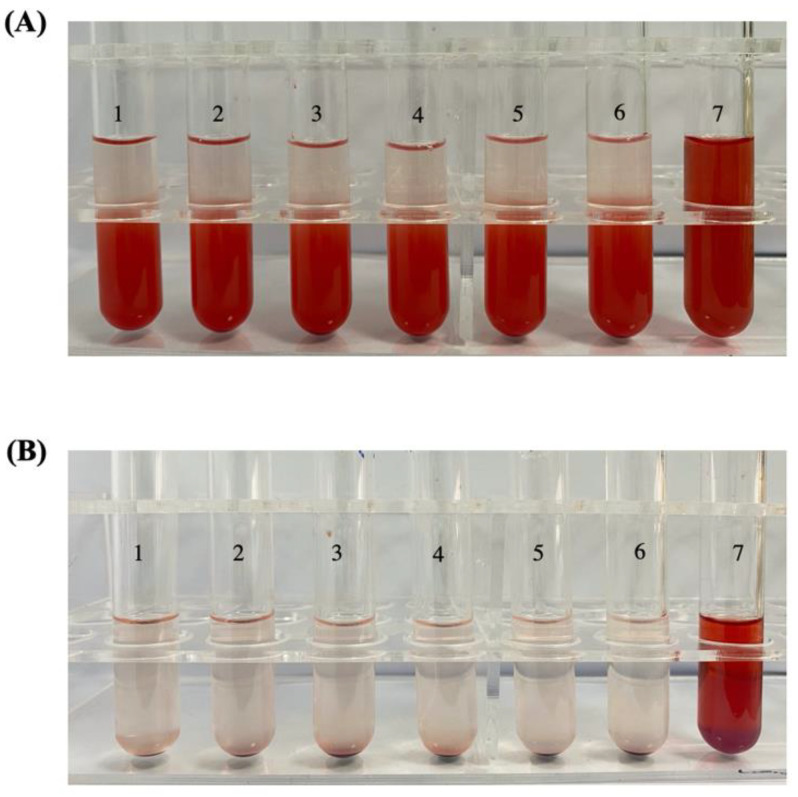
Hemolysis result of RBCs treated with RPTR-701Ns for; (**A**) 3 h and (**B**) 24 h.

**Figure 6 pharmaceutics-13-00099-f006:**
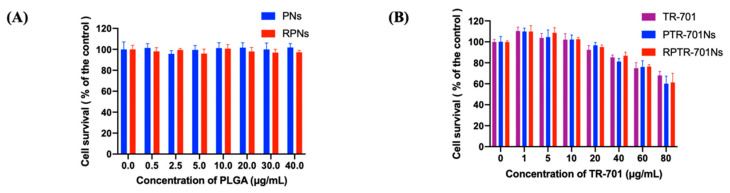
Cell survival ratio of HEK 293 cells after treated with; (**A**) nanoparticle material; and (**B**) drug loading nanoparticles with different concentration for 24 h. Data were presented as the mean ± SD (*n* = 6).

**Figure 7 pharmaceutics-13-00099-f007:**
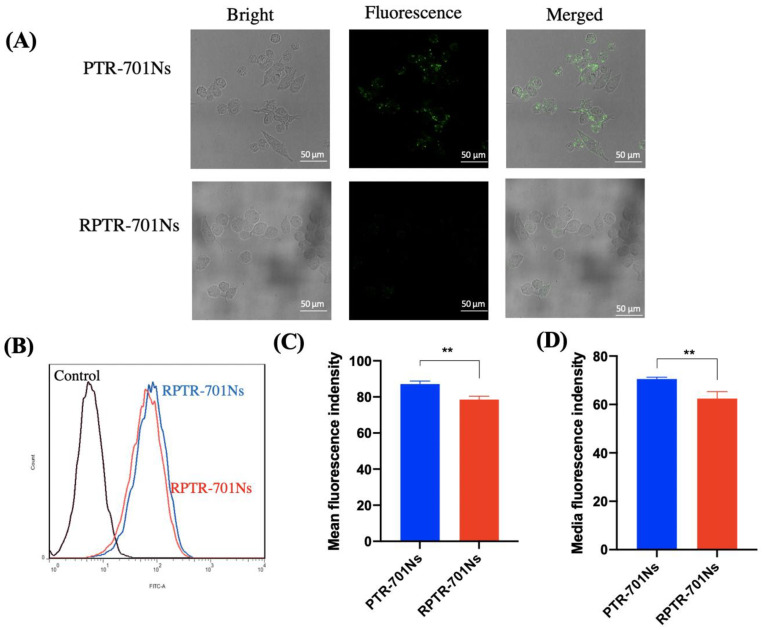
Immune-escaping capability of RPTR-701Ns. (**A**) LSCM images of uptake of PTR-701Ns and RPTR-701Ns by RAW264.7 cells. PLGA was labelled with DiO (green). (**B**) Fluorescence intensity was examined by a flow cytometer. (**C**) Quantitative analysis of the mean and (**D**) Media fluorescence intensity. Data were presented as mean ± SD (*n* = 3), ** *p* < 0.01.

**Figure 8 pharmaceutics-13-00099-f008:**
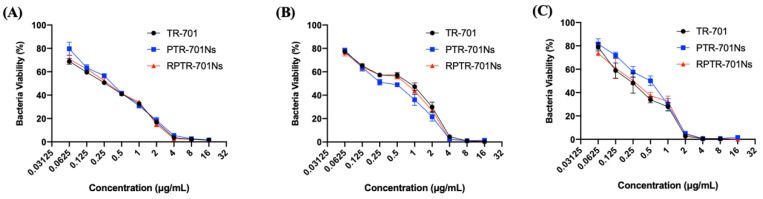
In vitro anti-bacteria efficacy of nanoparticles against, (**A**) MRSA USA 300, (**B**) MRSA ATCC 43300 and (**C**) MSSA NTCT 8325. Bacteria were administrated with TR-701, PTR-701Ns and RPTR-701Ns at different TR-701 concentrations. Data were presented as mean ± SD (*n* = 6).

**Figure 9 pharmaceutics-13-00099-f009:**
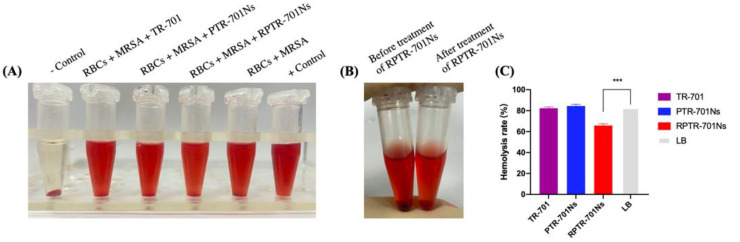
Exotoxin-removing capability of RPTR-701Ns. (**A**) Results of the anti-hemolysis of RPTR-701Ns to exotoxins secreted different kinds of bacteria. Luria Bertani (LB) solution was set as the negative control and 0.1% Triton X-100 LB solution was set as the positive control. (**B**) RBCs treated before and after RPTR-701Ns and (**C**) Quantitative analysis of anti-hemolysis. Data were presented as mean ± SD, *n* = 6, *** *p* < 0.001.

**Figure 10 pharmaceutics-13-00099-f010:**
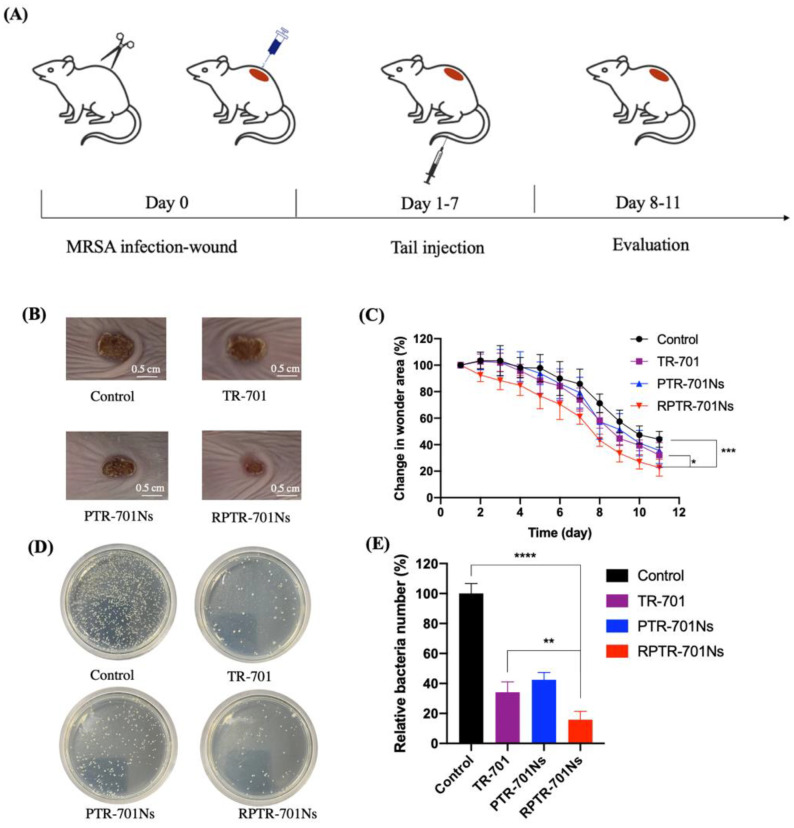
In vivo anti-bacterial efficacy of the formulations in MRSA-infected wound beading Balb/C mice. (**A**). Schematic diagram of MRSA-infected model establishment and treatment plan. (**B**) Photos of wound skin after treated with PBS, TR-701, PTR-701Ns and RPTR-701Ns. (**C**) Relative change in the wound area, *n* = 6. (**D**) LB culture pates of different groups of skin. (**E**) Quantitative analysis of bacteria colony, *n* = 4. (**F**) H&E staining of wound skin. Data were presented as mean ** *p* < 0.01, **** *p* < 0.0001.

**Figure 11 pharmaceutics-13-00099-f011:**
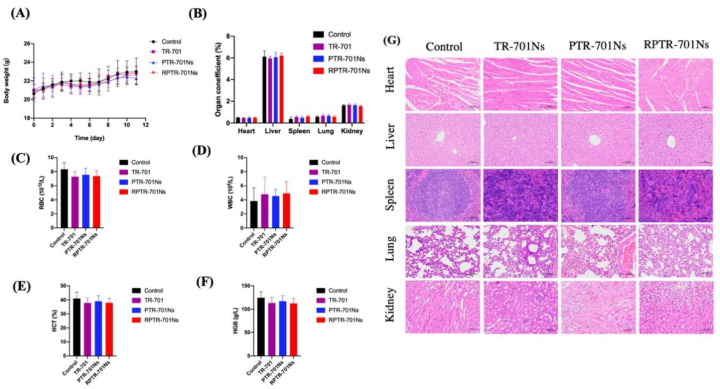
In vivo safety of the treatment. (**A**) Body weight changes in different groups during the treatment. (**B**) Organ coefficients. (**C**) RBCs, (**D**) WBCs, (**E**) HCT, (**F**) HGB, (**G)** H&E staining of main organs. Data were presented as mean SD (*n* = 6).

**Table 1 pharmaceutics-13-00099-t001:** Hemolysis Test of RPTR-701Ns.

Tube No.	1	2	3	4	5	6	7
Normal saline (mL)	2.4	2.3	2.2	2.1	2.0	2.5	0
Ultrapure water (mL)	0	0	0	0	0	0	2.5
2% RBCs (mL)	2.5	2.5	2.5	2.5	2.5	2.5	2.5
RPTR-701Ns (mL)	0.1	0.2	0.3	0.4	0.5	0	0

## Data Availability

The data presented in this study are available on request from the corresponding author.

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
