# Peer review of "Red Blood Cell Membrane-Camouflaged Tedizolid Phosphate-Loaded PLGA Nanoparticles for Bacterial-Infection Therapy"

_pharmaceutics, 2021, doi:10.3390/pharmaceutics13010099_

Round 1

Reviewer 1 Report

The reviewed manuscript presents the results of research on the formulation and application of the Tedizolid Phosphate release system, formed from PLGA nanoparticles coated by the membrane made from red blood cells. The work is interesting, showing both the method of creating a nanoparticle-based release system, a fairly new antibacterial compound recommended for the treatment of skin infections, as well as the preliminary results of in vitro and in vivo release kinetics of this drug. However, the purposefulness and effectiveness of using this system raise serious doubts.

Despite the fact that systems similar to the described systems based on nanoparticles of bioresorbable polymers or using RBCM are already known and attempts were made to apply them (https://doi.org/10.1016/j.apsb.2019.01.011), in my opinion, the work contains a relevant scientific novelty so that it can be published in Pharmaceutics.

However, in the manuscript I have noticed many passages  that must be refined and expanded. Only after clarifying a number of doubts which I have listed below, this work can be published.

Abstract

This part of the text is really hard to understand, it does not present the most important conclusions obtained from the research in a concentrated manner, but at the same time understandable for a potential reader. It does not at all encourage a study of the manuscript. The authors must thoroughly edit this text.

Introduction

The authors cite the relevant works, but for the most important ones, describing similar drug release systems, there is lack of deeper, more detailed description that would allow for a later comparison of the system developed by the authors with the similar and previously used ones.

In vitro release study

The lack of description of the course of HPLC measurements used (type of device, separation principle, measurement conditions, type of selected mobile phase, columns used, temperature, standards, etc.?

Preparation and Stability of PTR-701Ns and RPTR-701Ns

The purpose of the research described in this chapter is not entirely clear, as is the final conclusion source "It suggested that 37 ° C may be a better storage condition."

Drug Loading Capacity and In Vitro Release Study

The presented relationship between the time / cumulative amount of released drug proves a very fast release of tedizolite phosphate (half of the content within 1-2 h, practically all the drug is released after 10 hours). So, there is no prolonged-release effect of the maintenance dose. What is the point of using such a complex system to release this drug then? After all, you can achieve a similar effect in a much simpler way.

In Vitro Antibacterial Efficacy Study

The conducted research described in this section rather, they bring nothing. By knowing the weight content of the drug in the formulated system, the drug release rate, and the drug MIC, an effective dose value can be calculated. The time of full and partial antimicrobial activity of the produced system should bee  more interesting than that of the drug itself. There is no such research here.

Determining the length of time the system is antimicrobials active compared to the same dose of "naked" drug would be much more interesting.

In Vivo Ant-infection Study

There is lack of the deeper comment. The dry results suggest that the application of the described release system practically does not change anything, so I conclude that it is redundant. Am I right?

Discussion

This part requires a thorough development. The authors should clarify the   signalized doubts. They should to demonstrate the advantages and disadvantages of the proposed system against the background of similar systems described earlier. The authors should also present their suggestions related to possible possibilities of practical application of the described solution.

Author Response

Dear Reviewer:

We deeply appreciate the time and efforts you’ve spent in reviewing our manuscript and we thank you for your comments. Those comments are all valuable and very helpful for revising and improving our paper, as well as the important guide to our researches. We have studied the comments carefully and have made some changes and reply which we hope address the comments. We have marked the changes in track-change mode in the latest version of manuscript. The responses to the comments are attached.

Reviewer 2 Report

Please, find the comments in the attached file

Author Response

(The authors gave the same response as above.)

Round 2

Reviewer 1 Report

The authors have responded fairly fully to most of my questions and comments. Appropriate amendments to the text were also introduced. Two issues, however, have been quite marginalized.

Question 6.
I understand the authors' answer but, I completely disagree with the thesis presented that the effectiveness of an antibacterial system is determined only by the achievement of complete elimination of bacteria in the shortest possible stage of system application. Many bacterial infections (especially of the skin or mouth) are repeated infections, renewing themselves when the concentration or activity of the antibacterial agent used is sufficiently decreased. Rather, in such systems, it is necessary to maintain a locally sufficiently high concentration of the drug over an extended period of time. This phenomenon also prevents the formation of bacterial strains showing acquired drug resistance. Such studies have been conducted and described in the literature. The use of release systems is one of the reasons for the increased efficiency of the drug release system compared to "naked" drug. The authors observed this themselves and commented in the next answer.

Question 8.
The authors did not propose examples of the specific therapies, where treatment with the proposed system could be more effective than current used.

Author Response

Dear Reviewer:

We deeply appreciate the time and efforts you’ve spent in reviewing our manuscript and we thank you for your comments. We have studied the comments carefully and have made changes which we hope address the comments. The main changes are marked in track-change mode in the revised manuscript. The responses to the comments are attached.

Reviewer 2 Report

Authors defined the preperation of the TR-701 solution as: 30 mg/mL TR-701 was dissolved with 30 mg/mL NaHCO3. Again, this sentence in NOT clear. From what I understand, authors have a solution of TR-701 with a concentration of 30 mg/mL (in which solvent?) and dissolve it in a solution of NaHCO3 with a concentration of 30 mg/mL (if authors use a solution of NaHCO3, authors have to give the concentration of this solution), therefore the final TR-701 concentration is 30 mg/mL (but again, is also 30 mg/mL the initial concentration of the NaHCO3?). The experimental section have to be clear enough for other researchers to repeat authors’ experiments. As it is right now, it is not posible to understand completely the experimental process conducted to prepare the TR-701 solution

Author Response

Dear Reviewer:

We deeply appreciate the time and efforts you’ve spent in reviewing our manuscript and we thank you for your comments. We have studied the comment carefully and have made changes which we hope address the comment. The main changes are marked in track-change mode in the revised manuscript. The responses to the comment is attached.
